# Mindful Parenting Intervention MinUTo App for Parents of Preschool Children: Study Protocol of a Randomised Controlled Trial

**DOI:** 10.3390/ijerph19137564

**Published:** 2022-06-21

**Authors:** Annalisa Guarini, Alessandra Sansavini, Chiara Suttora, Stefania Bortolotti, Margherita Fort, Daniela Iorio, Chiara Monfardini, Maria Bigoni

**Affiliations:** 1Department of Psychology “Renzo Canestrari”, University of Bologna, 40126 Bologna, Italy; alessandra.sansavini@unibo.it (A.S.); chiara.suttora@unibo.it (C.S.); 2Department of Economics, University of Bologna, 40126 Bologna, Italy; stefania.bortolotti@unibo.it (S.B.); margherita.fort@unibo.it (M.F.); daniela.iorio@unibo.it (D.I.); chiara.monfardini@unibo.it (C.M.)

**Keywords:** mindful parenting, APP, preschool children, randomised controlled trial, study protocol

## Abstract

Background: Mindful parenting and the use of technology for parenting intervention have expanded separately from one another with promising results, but their relationship is underexplored. The current study protocol proposes a new universal intervention via app, MINd Us TOghether (MinUTo), based on mindful parenting for parents of typically developing children of 4–5 years of age. Methods: The effect of the intervention is evaluated using a randomised controlled trial. Around 2000 parents are enrolled and randomised to the intervention and control groups. Data are collected in three different waves from parents at baseline and endline; APP usage data allow for the analysis of intervention adherence. The MinUTo app proposes contents and activities for five dimensions of mindful parenting. Each dimension is presented within a two-week distance, explaining its importance, providing information, and offering activities for parents and children. Expected results: We hypothesise a positive effect of the intervention on primary outcomes (mindful parenting, parenting stress, parent behaviours and parental time investment), increasing parents’ skills and promoting a positive parent–child relationship. We also test possible effects on secondary outcomes (parenting attitudes and beliefs) at an explorative level. Conclusions: The study will add new considerations about the psychological and economic impact of technologies in implementing parenting interventions in non-clinical populations.

## 1. Introduction

### 1.1. Mindful Parenting

Several authors defined mindfulness as the intentional non-judgmental awareness of the present moment [1]. The construct of mindfulness was associated with parenting for the first time by Kabat-Zinn and Kabat-Zinn [2] with the concept of mindful parenting. A recent meta-analysis described the relationship between parent mindfulness and mindful parenting, suggesting an overlap between the two measures as parents with a high level of mindfulness can apply their skills in parenting [3]. Recently, mindful parenting was defined not “as a project to create “better” or “optimal” children [..] or to be “better” or “optimal” parents, but to embrace in moment-to-moment awareness as best we might the entire enterprise of parenting our children in a mutuality of love and discovery and not-knowing” (p. 266) [4]. In addition, the authors highlighted that mindful parenting is an ongoing process in each moment instead of an endpoint.

The first suggestions for fostering mindfulness in parenting training were described as strategies for overcoming the grip of automaticity in families with disruptive children through facilitative listening, distancing, and motivated action plans [5]. A detailed description of skills included in the mindful parenting concept was done by Duncan and colleagues [6]. The authors suggested five skills: listening with full attention, non-judgmental acceptance of self and child, emotional awareness of self and child, self-regulation in the parenting relationship and compassion for self and child. Similar skills were described by Kabat-Zinn and Kabat-Zinn [4] including awareness of child’s feelings and needs, listening with full attention, acceptance of pleasant or unpleasant things, recognizing reactive impulses and learning to respond.

### 1.2. Mindful Parenting Interventions

Mindful parenting interventions have been proposed to support parents of children and adolescents with typical, at-risk, and atypical development, improving intrapersonal and interpersonal outcomes [7]. The following paragraph mainly summarises the findings of reviews and meta-analyses.

Several experiences of mindful parenting training and interventions have been proposed for parents of children at risk or with atypical development. Mindfulness training has been offered to parents of children with ADHD, showing positive effects in reducing parenting stress and children’s symptoms [8]. Positive effects of mindfulness interventions for parents of children between 3 and 12 years with externalising behaviour were described with decreased parental stress and child problems [9]. Mindful parenting also showed promising results in improving the well-being of parents of children with autism spectrum disorder [10]. In addition, the positive effects of mindful parenting on parents’ well-being were more evident in parents of children with medical than psychological conditions [7]. However, despite the promising results of mindfulness-based interventions in promoting parents’ mental health of children with developmental disorders, these reviews and meta-analyses suggested caution due to the small number of studies carried out [8]. Evidence, indeed, is still at an early stage, calling for more randomised controlled trials to consolidate these findings in the literature [11].

For what concerns mindful parenting programs applied to typically developing samples, relatively limited studies with mixed results have been described [12,13]. Mindful parenting interventions have been proposed to parents at different times of their experience. Indeed, a recent review suggested that these interventions promote parenting skills in the postpartum period and parent-infant relationships in the early years [14]. Mindful parenting interventions have also been proposed to parents of children and adolescents, reducing parenting stress with a slight improvement in youth outcomes [15]. However, more research on non-clinical populations is needed to understand the effect of these programs as preventative benefits [12,13].

### 1.3. Mindful Parenting Interventions Using APPs

In the digital age, parents use apps to find information about health and child development, learn about milestones, and find activities to support the parent–child relationship. However, the abundance of low-quality apps is dangerous for parents in finding the correct information [16]. Indeed, in the study of Virani and colleagues [16], 4300 apps were identified, but only 16 (0.4%) met the quality criteria. Following these considerations, in the present paragraph we focus on evidence-based apps.

Some promising experiences of app use were described to support early parenting, i.e., “Growing Together” [17] and “BabyMind” [18]. Technologies were also used to help parents in promoting eating and physical activities (app, [19]) or children’s academic development (text-messaging program READY4K!, in 4-year-old children [20]). Apps have also been proposed to increase safety knowledge and actions to prevent unintentional child injuries [21]. Parenting apps had given precious support during the COVID-19 pandemic when a reduction in health care and social assistance occurred [22].

Concerning mindful parenting interventions proposed via apps, only two experiences have been described. The first mobile-based intervention was offered for parents of children (8–18 years) with chronic pain. Findings suggested a decrease in parental stress and solicitous behaviours and an increase in mindful parenting, suggesting the possibility of delivering intervention through a mobile device [23]. A second mindful digital intervention, Baby Care Program, was proposed for pregnant women. Results were promising, with a decrease in stress levels in mothers and an improvement in sleep quality and nutrition control. In addition, positive correlations between app use and childhood development and milestones have been described [24].

To our knowledge, mindful parenting apps for parents of children at preschool age with typical development have not been proposed using an evidence-based approach. Promising results in the last ten years have been described by other interventions to support parents of children aged 4–5 years, confirming the relevance of helping parents of children at preschool age. For example, Tuning in to Kids is an intervention encouraging changes in parenting beliefs and behaviours, increasing the emotional connection in parent–child relationships [25].

### 1.4. Aims

As recently suggested [26], mindful parenting and the use of technology for parenting interventions have expanded largely separately from one another. The relationship between mindful parenting and technology use is underexplored and complex. In addition, this relationship in mindful parenting of children of 4–5 years is entirely lacking.

We propose the intervention MINd Us TOghether (MinUTo) with the following characteristics. First, the MinUTo intervention is based on mindful parenting using Duncan’s model [6] and includes the five proposed dimensions (listening with full attention, non-judgmental acceptance of self and child, emotional awareness of self and child, self-regulation in the parenting relationship, compassion for self and child). We use Duncan’s model for its completeness and because it is well-recognised at the international level (364 citations May 2022; Scopus database).

Second, the intervention is distributed via app, a feature that will allow for the scale-up of the intervention at a relatively low cost. Indeed, more research on mindful parenting training is needed in non-clinical populations, including community samples, to test possible preventative benefits [12,27]. Following Virani and colleagues’ suggestion [16], the app MinUTo is developed with a strategic partnership between academic researchers and a commercial company to create an app. As a consequence, it integrates two attractive features: it is scientifically founded and appealing to users. In the end, as we know that parents’ smartphone use can affect the parent–child relationship [28], we suggest that parents use the app MinUTo in a quiet moment, not during parent–child interaction.

The third characteristic of the MinUTo intervention is the universal design, such as in other universal parenting interventions [29]. The advantage of universal interventions is that all parents in a specific geographic area are invited, avoiding labelling or stigmatization [30]. Notwithstanding our efforts to reach a diverse population of parents, given the substantial investment expected from parents in implementing the suggestions offered via MinUTo, we maintain that participation to MinUTo is voluntary.

Fourth, the intervention proposes both activities to be carried out by parents alone and activities that involve parent–child interaction (i.e., book sharing). A potential advantage was described for intervention programs where both parents and children have been involved [7].

Lastly, as several studies suggested the need for greater methodological rigour in evaluations of mindfulness-based family interventions [27], we propose a randomised controlled trial (RCT).

The present study aims at analysing the effectiveness of MinUTo intervention on:(a)Increasing mindful parenting (primary outcome);(b)Decreasing parenting stress (primary outcome);(c)Increasing functional and adaptive parent behaviours (primary outcome);(d)Increasing the quality of parental time investment with the child (primary outcome);(e)Changing parenting attitudes (secondary outcome);(f)Changing parenting beliefs (secondary outcome).

## 2. Methods/Design

### 2.1. Eligibility Criteria

Eligible participants are parents of children aged 4- or 5-years attending a municipal kindergarten in the Emilia-Romagna Region. Eligible participants in the pilot study, wave 1 (2020–2021), were all parents of children attending municipal kindergarten in Bologna. Eligible participants in wave 2 (2021–2022) were parents of children attending municipal kindergartens in Bologna and Ravenna. In the third wave of data collection (2022–2023), we plan to extend the sample to families of children attending kindergartners in other municipalities of the Emilia-Romagna region.

Participation is completely free of charge and voluntary. To increase the program take-up and response rates to surveys aimed at collecting the necessary data sources for the MinUTo program evaluation, we administer the intervention and related questionnaires in Italian, English and French and offer a monetary reward to the children’s schools, proportional to the number of parents who complete the baseline and endline questionnaires.

### 2.2. Randomisation

A meta-analysis suggests that parenting intervention may have moderate size effects [7] in a non-clinical sample. We ask all parents to fill in baseline questionnaires. To improve estimates precision, we reviewed the relevant literature and selected variables correlated with the main outcome considered in the study. We stratify the sample by considering child’s cohort of birth, presence of siblings, and indicators of parents’ employment and education. Within each stratum, parents are randomly assigned to one of the treatment arms: intervention and control.

All parents are asked to fill in the endline assessment. However, parents assigned to the intervention group receive the app experience, while parents in the control group do not receive the app experience between baseline and endline assessments, but after endline data collection. It is important to note that parents are not notified about the assigned treatment arm until the end of the baseline survey compilation period (Figure 1).

### 2.3. Ethics

The study protocol met the ethical guidelines for the protection of human participants and received formal approval from the local Ethical Committee (Bioethics Committee, University of Bologna, Bologna, Italy). The parents release their informed written consent for participation in the study and data protection according to the information provided by the research team.

### 2.4. Tools

Data are collected at two points: baseline and endline (Figure 1). Five self-reported questionnaires and two-time use surveys are provided to parents at baseline and again, after the completion of the intervention.

#### 2.4.1. Interpersonal Mindfulness IM-P

To assess mindful parenting, the Interpersonal Mindfulness in Parenting (IM-P) [31] is proposed to parents. The IM-P questionnaire has been developed and validated in Dutch [6,32] and intends to assess mindfulness under the interpersonal domain of parent–child relationships. It comprises 31 items covering the following five subscales: (1) Listening with full attention refers to the parental ability to listen to the child with focused attention and awareness (five items, Cronbach’s alpha = 0.83); (2) Emotional awareness of self and child regards the parental ability to of being aware of his/her own and child’s emotions (six items, Cronbach’s alpha = 0.54); (3) Self-regulation in the parenting relationship refers to the parental ability to be less reactive and more regulated in response to child’s behaviour (six items, Cronbach’s alpha = 0.76); (4) Non-judgmental acceptance of self and child concerns the parental awareness of the expectations on the child and the acceptance of the child’s traits and behaviours (seven items, Cronbach’s alpha = 0.74); (5) Compassion for self and child refers to developing a genuine stance of caring and compassion for their child as well as for themselves as parents (seven items, Cronbach’s alpha = 0.78). Parents are asked to judge the items with a Likert scale ranging from 1 (never true) to 5 (always true); therefore, higher scores reflect a higher level of mindfulness in parenting. The questionnaire has been translated into Italian by the research group using a back-translation method.

#### 2.4.2. Parenting Stress Index-Short Form PSI-SF

Parenting Stress Index Short Form (PSI-SF [33], Italian version [34]) is a commonly used measure of stress related to the parenting role. The questionnaire includes 36 items and consists of the following subscales: Parental Distress refers to the level of distress the parent is experiencing in his/her parental role; Parent–child Dysfunctional Interaction assesses the parent’s perception that the child is not meeting the parental expectations and that parent–child interaction is not reinforcing; Difficult Child regards the parent’s view of the child’s self-regulatory abilities; Total Stress represents a global score of parental distress given by the sum of all items. The PSI–SF also contains a Defensive Responding scale indicating the degree to which the parent might be attempting to deny parenting problems. Items are rated using a 5-point Likert scale (1 strongly disagree; 5 strongly agree); high values indicate a high level of parenting stress. PSI-SF global scale and subscales showed high internal consistency in the Italian validation [34]; Cronbach’s alphas were: 0.80 for Parental Distress, 0.81 for Parent–child Dysfunctional Interaction, 0.72 for Difficult Child, 0.89 for Total Stress; 0.70 for Defensive Responding.

#### 2.4.3. Parent Behaviour CECPAQ

The Comprehensive Early Childhood Parenting Questionnaire (CECPAQ) [35] is a 54-item questionnaire developed for Dutch parents of pre-schoolers and assessing parenting in its multidimensional nature. Based on three theories on parenting (attachment theory, Vygotsky’s sociocultural theory of learning and Bandura’s social learning theory), the questionnaire measures commonly occurring behaviours in five macro-dimensions of parenting, which are central to early childhood, i.e., Support, Stimulation, Structure, Harsh Discipline, Positive Discipline. Parents are asked to rate their parenting behaviours on a 6-point scale, ranging from 1 (never) to 6 (always), for most of the items included in the scale. However, for nine items (items 14–23), parents are asked to use a 6-point scale anchored on one effective and one ineffective response to the parenting situation (e.g., “When there is a problem with my child” from 1: “Things build up and I do things I don’t mean” to 6 “Things don’t get out of hand”).

The CECPAQ five-factor structure was confirmed for the Dutch [35] and the Chinese versions [36] and reliability was also corroborated. For the Dutch sample, Cronbach’s alphas for mothers and fathers were high and were: 0.88/0.88 for Support, 0.82/0.86 for Stimulation, 0.75/0.77 for Structure, 0.79/0.79 for Harsh Discipline, and 0.76/0.77 for Positive Discipline, respectively. The questionnaire has been translated into Italian by the researchers involved in the present study through the method of back translation.

#### 2.4.4. Time Use

Information about time investments of parents is collected through time diaries, which provide a more accurate and reliable measure of time allocation with respect to that obtained through retrospective questionnaires [37]. For this purpose, the research group developed a new app-based time use survey following the Eurostat guidelines [38] to identify the macro-categories and the activities to be included in the questionnaire. We also rely on encouraging results from recent experiences of time use diary collection through apps using mobile devices [39,40]. Parents are asked to describe their time use over the whole 24 h spectrum of two reference days (one weekday and one weekend day—randomly assigned), selecting from a list of 137 activities divided into 11 macro-categories. For each episode, the time use survey records the duration of the activity (with a minimum length of 10 min) and the social context, i.e., with whom the activity is carried out. Consistently with the main purpose of the MinUTo program, when the child is present, the parent declares whether and how much the child is involved in the activity. For activities such as childcare that entail an obvious involvement of the child, this step is skipped. If the child is involved, information on parent’s and child’s mood during the activity episode is asked (cf. the Well Being Module of the American Time Use Survey, ATUS, U.S. Bureau of Labor Statistics, Washington, DC, USA). This makes it possible to measure time investments of the parent not only as the time spent with the child in total or in a given set of activities, but as the time during which there is parent–child interaction, and during which the child feels positive emotions. More specifically, we define four measures of parental time investments: (i) the total time spent in the presence of the child; (ii) the quality time as defined by Price [41], which includes a set of activities that are expected to involve a high degree of parent–child interaction and to have the greatest impact on child development (reading, doing homework, doing arts and crafts, doing sport, playing, attending performances and museums, engaging in religious activity, having meals and talking with the child, or providing personal care for the child); (iii) the time spent with the child actively involved regardless the kind of activity performed; (iv) the time spent with the child actively involved and feeling happy. Previous work covering indicators of happiness and quality time related to the project appeared in the master thesis of Forti [42] and Sansone [43].

#### 2.4.5. Parenting Attitudes EPAQ

The Early Parenting Attitudes Questionnaire (EPAQ; [44]) has been developed to assess parents’ variation in intuitive theories of parenting. It includes 24 items equally divided into the following three scales: (1) Affection and Attachment evaluates the idea that emotionally close parent–child relationships are relevant for development; (2) Early Learning scale assesses parents’ attitudes toward the importance of fostering early learning; (3) Rules and Respect scale investigates parental attitudes toward the relevance of rules and respect. Parents are asked to use a 7-point Likert scale to report how much they agree upon a series of statements from 0 (do not agree) to 6 (strongly agree) (e.g., “Children should be comforted when they are scared or unhappy”). Reported reliability documented a Cronbach’s alpha for the whole scale of 0.90 and 0.82, 0.83, and 0.81, respectively, for the subscales. The questionnaire has been translated into Italian by the researchers involved in the present study through the method of back translation.

#### 2.4.6. Parenting Beliefs

This survey has been built by the research team for the purposes of this project to elicit parents’ beliefs on how much their own involvement in the activities carried out by the child is important for the child’s wellbeing, and whether parents have different beliefs depending on the specific activities performed with the child [45]. The questionnaire includes 44 paired items: each pair of items concerns a specific activity, which the child can carry out together with the parent, or without the parent but possibly with other children or adults, and it is directed to assess to what extent parents believe the activity is beneficial or harmful for a child’s wellbeing, under the two alternative scenarios (e.g., “The child reads/looks through a book together with his/her parents” and “The child reads/looks through a book alone or with other children”). Parents are asked to rate each item on a slider ranging from “Reduces wellbeing” (coded as −100) to “Promotes wellbeing” (coded as +100). The first measure of parenting beliefs is based on the difference between the rating of the paired items, which is on average larger for parents who think that their own engagement with the child is particularly beneficial [46]. The questionnaire also allows us to identify the subset of activities that the parents consider more beneficial for the child when performed with parental involvement and the subset of activities where parental involvement may be detrimental compared to performing activities alone or with peers. We also plan to explore the underlying factor structure [45].

## 3. Intervention Program

In the introduction, we described the model of mindful parenting elaborated by Duncan and colleagues [6] and its components. This model constitutes the theoretical basis of the online intervention, as it aims at enhancing parents’ awareness of their parental role and their relationships with their children. As already discussed, Duncan’s mindful parenting model encompasses five relevant dimensions inspired by the more general theory of Mindfulness. These are: (a) listening with full attention; (b) non-judgmental acceptance of self and child; (c) emotional awareness of self and child; (d) self-regulation in the parenting relationship; (e) compassion for self and child. Listening with full attention is a core dimension of mindfulness [47] and its relevance is particularly evident both at a personal and an interpersonal level. With reference to parenting, this skill supports parents in connecting with their children, helping them to show a high sensitivity to their emotional needs and fully understand their verbal and non-verbal communicative signals. The second dimension—non-judgmental acceptance of self and child—refers to a non-judgmental attitude towards oneself and other inner experiences, leading the person to lower his/her defences to negative emotions and states and accept these experiences. This attitude helps parents free themselves from biased beliefs and subconscious judgments about their child’s attributes and competencies and parental role within parent–child relationships. Emotional awareness of self and child regards parental capacity to identify emotions within themselves and their child correctly. The ability to regulate reactivity to strong emotions in the parent–child relationship is captured by the fourth dimension, i.e., self-regulation in the parenting relationship. Parents who can master this skill can help their children to develop their self-regulation abilities. Finally, the dimension compassion for self and child implies an emphatic concern for the child and oneself as a parent, accompanied by the desire to alleviate the suffering.

The online intervention is provided to parents via an app called MinUTo, downloadable on various devices such as smartphones, tablets, and desktop applications. After registration, which includes informed consent, a brief survey with questions regarding parents’ and children’s socio-demographic information and baseline assessment (see tools section), parents are provided with app experiences. The MinUTo app is developed by Indici Opponibili (https://www.indiciopponibili.com, accessed on 22 May 2022), designing a parents’ online experience to be enjoyable, inclusive and immersive [48]. Experiences were divided into five steps, each focused on a single model dimension; the app proposes each dimension with a two-week distance.

For each experience, three app areas are offered with a fixed order of appearance. The first one is called “Why is this important?” (Figure 2).

Within this first area, parents can find three reasons for increasing each specific dimension of their parenting role and the child’s wellbeing and development. For instance, listening with full attention helps parents: (1) to understand the child’s verbal and non-verbal communication (Figure 2); (2) to assess the child’s emotions and understand their impact on the parent–child relationship; (3) to reconsider their expectations and desires and listen to the child.

Once the parent ends exploring this area, a second one within the same dimension is activated. This is called “Things to know” (Figure 3) and includes three online interactive activities for parents to increase that specific dimension. These activities are provided in various formats—i.e., brief stories and games requiring the parent to interact with the app. For example, for the listening with full attention dimension, one activity is called “Listening is not the same as hearing!” (Figure 4a). Here, parents are called to listen to a sound wave (e.g., a nature soundtrack), trying to free their minds and focus only on the listening experience. Another activity included in the first dimension is called “Your child is like an orchestra” (Figure 4b); here, an illustration of a child is provided and by clicking on different body areas, such as the child’s hands, parents discover how the child communicates with this mean and how it is essential to focus on these non-verbal signals to capture the child’s communication in its wholeness and complexity.

After these three interactive online activities are concluded, the app shows parents the last area, “Things to do” (Figure 3), which includes two activities proposed within a week lag, developed to practice that specific dimension with his/her child. Concerning listening with full attention dimension, two activities are suggested. The first, “Tell me about yourself” (Figure 5), asks parents to take some time during the week to tell first about a positive event of their day, later about a negative one, to the child and to ask the child to do the same. Parents should try to apply to this exercise what they have learned since then, to give the child’s narration their full attention.

To support parents in proposing activities, we provide suggestions and reinforcements to avoid feeling unsuccessful about the realisation of the activity proposed. For the activity “Tell me about yourself”, parents received some tips including: “You were thinking of something else and get the impression you didn’t fully capture your child’s communication. Try and think of what was concerning you most at that time and try finding more space for next time”.

### Implementation Fidelity

Our measure of fidelity is given by the number of experiences completed by each treated parent, ranging from 0—for parents who never actually use the app, even if they have access to it—to 5 for parents who completed the whole program. 

## 4. Outcomes

### 4.1. Primary Outcomes

(a)To increase mindful parenting. We hypothesise a direct effect of the MinUTo intervention on mindful parenting, measured through the IM-P self-report questionnaire [31]. Other studies have described the direct impact of mindful interventions on mindful parenting [49]. However, the present study adds the analysis of the dimensions of Duncan’s model. We hypothesise that all the dimensions (i.e., Listening with full attention; Emotional awareness of self and child; Self-regulation in the parenting relationship; Non-judgmental acceptance of self and child; Compassion for self and child) can be positively affected. Indeed, the MinUTo intervention proposes materials and activities for each dimension.(b)To reduce parenting stress. We also hypothesise a direct effect of the MinUTo intervention on parenting stress, measured in the present study using the Parenting Stress Index Short Form (PSI-SF Italian version [34]). A direct effect of mindful intervention in reducing parenting stress was described in several studies considering parents of children and adolescents with typical and atypical development [8,15]. In our study, we hypothesise a positive effect on the Total Stress global score as well as on Parental Distress and Parent–child Dysfunctional Interaction subscales. Indeed, mindful activities proposed in the MinUTo intervention can support parents to feel competent in their roles and more satisfied in the interactions with their children. By contrast, we do not expect a significant effect in the Difficult Child subscale. Indeed, the proposed activities do not directly offer strategies to handle behaviour or support parents in perceiving their children as easy or challenging to take care of. No significant effect of the intervention is expected in the Defensive Responding subscale as it does not aim to reduce or increase defensive answers (e.g., trying to minimise problems). Our hypothesis, aligned with Shorey and colleagues [13], is to consider parenting stress not as a whole, taking into account that mindful parenting interventions may be more effective in reducing specific components of parenting stress.(c)To increase appropriate and functional parent behaviour. We assess parent behaviour using the CECPAQ questionnaire [35]. We hypothesise that the MinUTo intervention can increase appropriate and functional parent behaviour, according to previous studies [50]. For example, promoting self-compassion allows parents to increase their parental presence, breaking the cycle of repetitive and negative thoughts [50]. We also hypothesise a significant effect on all subscales, with an increase of sensitivity, responsiveness and affection of parents, more activities and toys proposed, and a higher consistency with a reduction of over-reactivity or laxness, reducing physically, verbally, and psychologically aggressive punishment. Indeed, as suggested by the literature [13], mindful parenting training can reduce parental automatised behavioural responses and allow parents to reflect and choose appropriate and functional behaviours.(d)To increase the quality of parental time investment with the child. Thanks to the time survey, we collect information about the quality of the daily life parent–child interaction. We hypothesise that the MinUTo intervention can affect parental time investments by increasing the quality time as defined by Price [41]; the time with the child actively involved; the time with the child actively involved and feeling happy. Indeed, our intervention focuses on supporting parents in understanding and practising being present and aware in everyday interactions with children without judgment in mutuality of love and discovery [4]. By contrast, we do not hypothesise that MinUTo intervention can increase the total time spent with the child, as external constraints can limit this possibility (e.g., hours of work; family compositions; care of relatives, etc).

### 4.2. Secondary Outcomes

(a)To change parenting attitudes. The Early Parenting Attitudes Questionnaire (EPAQ; [44]) is included in the questionnaires administered to parents in the pre- and post-intervention assessments. We hypothesise a possible effect in the Affection and Attachment subscale mediated by interpersonal mindfulness in parenting. Indeed, we hypothesise that parents with high mindful parenting skills can get closer to the idea that emotionally close parent–child relationships are relevant for development. By contrast, we do not expect differences in Early Learning and Rules and Respect subscales as we do not propose activities to improve parents’ attitudes toward the importance of fostering early learning and rules and respect.(b)To change parenting beliefs. We hypothesise an effect of the intervention of mindful parenting in strengthening parental beliefs on the importance of parental engagement in daily activities, as measured by the questionnaire we specifically designed for this purpose. The recent literature [51,52] illustrates how parental beliefs can respond to interventions in disadvantaged socioeconomic contexts. To our knowledge, our study would be the first to investigate the impact of a parenting program on parental beliefs in a developed country.

## 5. Statistical Analysis

### 5.1. Sample Size

Following the suggestions of Athey and Imbens [53], we consider a stratified RCT design as this approach is preferable to parametric regression adjustments for covariates. At the same time, the authors warn against the risk of using small strata and recommend avoiding pairing units. Building on these indications, we define strata based on child’s cohorts, presence of siblings, parental education, and employment status. Our strata definition takes into account the presence of missing data on any of these covariates and trades-off the requirement of a minimum sample size per stratum, with a meaningful definition of mutually exclusive strata. We aim to reach about 2000 participating parents, but we also anticipate a (conservative) drop of about 50% in endline response rates, as attrition is typically high in parental involvement studies [54]. As a consequence, with about 1000 observations at the endline for each outcome, the minimum detectable effect size, with a bilateral test contrasting parents randomly assigned to the intervention and control groups (at significance level alpha equal to 5% and a power of 80%) for a selected set of primary outcomes examined independently, is about 0.2 of a standard deviation for the general scale of CECPAQ and Interpersonal Mindfulness IM-P, as well as for total parenting stress of PSI, and for the time with the child actively involved. However, these figures are indicative; we plan to explore subscales and adjust the analysis for multiple hypothesis testing.

### 5.2. Planned Statistical Analysis

First, we will compute the internal consistency of subscales of questionnaires and surveys used in baseline and endline assessments to obtain a measure of reliability (Cronbach’s alpha). In addition, as some questionnaires (e.g., Interpersonal Mindfulness IM-P, Parenting behaviour CECPAQ, Parenting attitudes EPAQ) have not been validated in the Italian language and culture, we will also carry out Confirmatory Factor Analyses to confirm the factorial structure of the questionnaires and subscales.

Secondly, controlling for pre-existing differences, we will perform balancing tests, contrasting the intervention and control groups on the sociodemographic and baseline measures, i.e., Interpersonal Mindfulness IM-P, Parenting Stress Index PSI, Parenting behaviour CECPAQ, Time Use, Parenting attitudes EPAQ and Parenting beliefs. All these analyses, consistently with the stratified nature of our RCT design, will include strata fixed effects to capture the gains from stratification [53].

In the third step, we will run multiple regression analyses to assess the effect of the intervention on primary and secondary outcomes. As we will examine multiple outcomes separately, we also plan to perform a robustness analysis accounting for multiple hypothesis testing. We also plan to contrast results on the different outcomes, to get insights into the likely mechanisms generating the MinUTo impacts and the associations among variables. We expect significant non-compliance, namely some parents who are offered access to the MinUTo app do not use it. Control parents do not have direct access to the MinUTo app content and their identity is not known to other participating parents, thus we are confident they do not use the app. Considering the potential non-compliance of parents who were offered the app content, we focus our analysis on the effect of the offer of the app content on our primary and secondary outcomes.

Lastly, if an adequate sample size is achieved, our stratified RTC design will allow us to examine the heterogeneity of the effects across strata. We are specifically interested in the potential heterogeneity driven by child’s characteristics, parents’ education, and parental employment status, as well as those related to the presence of siblings.

## 6. Limitations

The present protocol shows some limitations. First, the efficacy of the intervention is evaluated using parents’ self-reported questionnaires. As suggested by Cohen and colleagues [55], very few studies have explored the effect of mindful parenting on child outcomes. Future studies should investigate the impact of MinUTo intervention on child psychological well-being to understand if modifications in parenting can have cascading effects on children. Child reports should be helpful to triangulate observations on interpersonal outcomes [13]. In addition, some questionnaires, even if widely used in other countries, are not validated for the Italian culture. We cannot exclude that they may show some limitations in the internal consistency and factorial structure.

Second, we collect data only at two points of observation (baseline before starting the intervention and endline at the end of the intervention). From one point of view, we cannot exclude that increased awareness due to the intervention could show worse scores at the endline while improving them in the long term. On the other hand, changes in behaviours documented after the intervention may be short-lived if they are not later reinforced and sustained. For these reasons, as suggested by other studies [13], a follow-up observation (after 6 or 12 months) will help to understand the long-term effects of MinUTo intervention on parents’ variables.

Third, the causal effect of the intervention will be mediated by the voluntary app use by treated participants. To minimize this issue, we collect baseline data to check the extent to which self-selection of parents may drive the results and we can assess the causal effect of being offered the MinUTo app on outcomes by comparing individuals who were randomly assigned to receive the app to those who were prevented from accessing the app content (control parents). We expect the two groups not to exhibit any statistically relevant difference at baseline, so that the comparison of endline outcomes between groups can be informative of the effect of having received the spp.

Lastly, we propose a universal intervention with several advantages, as described in the introduction. However, several disadvantages described in other studies can represent limitations [30]. It is more difficult to find overall effects in universal interventions than in targeted interventions. In addition, universal interventions might enlarge social inequality if primarily families with few risk factors voluntary participate in the interventions.

## 7. Conclusions

Mindful parenting programs applied to non-clinical samples have shown promising but contrasting results, suggesting the need for more research [12,13]. In addition, very few experiences of mindful parenting interventions proposed via apps have been described [23,24] and to our knowledge, mindful parenting apps for parents of children at preschool age with typical development have not been proposed using an evidence-based approach.

Starting from these considerations, the present study protocol aims at adding new evidence on the efficacy of mindful parenting via apps for parents of typically developing children at 4–5 years of age, proposing activities to be carried out by parents alone and with their children (i.e., book sharing). We use the RTC design to analyse the effectiveness of the MinUTo intervention with the methodological rigour needed in evaluations of mindfulness-based family interventions [27]. The MinUTo intervention is also based on a well-recognised model of mindful parenting [6]. Finally, we will propose the intervention with a universal design.

We hypothesise a positive effect on parents’ skills and the parent–child relationship as primary outcomes of the intervention. In particular, we expect an increase in mindful parenting, a decrease in parenting stress, an increase of adaptive parent behaviours and in the quality of parental time investment with the child. We also explore the effects of the intervention on parenting attitudes and beliefs as secondary outcomes, expecting a positive change in parenting attitudes and beliefs. If data collection and analyses confirm our hypothesis on the effectiveness of the MinUto intervention, several implications can be proposed.

As the app allows for the scale up of the intervention at a relatively low cost, we can propose the app to large numbers of families both at national and international levels according to the universal design of the current project. We are committed to release the app free of charge at the end of the project. The app has already been translated into three languages, but we can increase the number of languages to include parents from different nationalities. The MinUTo app can become a valuable and low-cost tool for the primary promotion of parenting and child wellbeing. As documented by the recent literature in the economics of education, a more intense parent–child interaction is associated with children’s socio-emotional development and, in particular, with their prosociality [56]. Socio-emotional skills have also been shown to represent key determinants of long-term outcomes in the child’s adult life [57,58]. For these reasons, we believe that an intervention fostering the quality of the parent–child relationship in the early years of life has the potential to contribute to the development of human capital for society at large.

## Figures and Tables

**Figure 1 ijerph-19-07564-f001:**
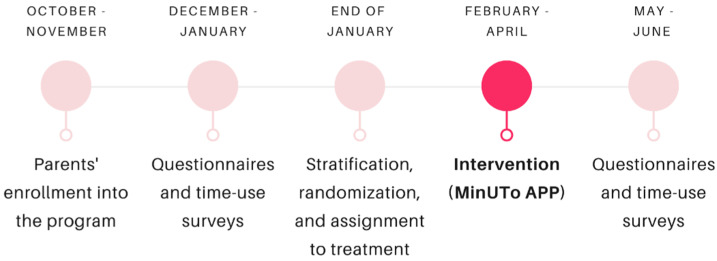
Timeline of MinUTo for each wave.

**Figure 2 ijerph-19-07564-f002:**
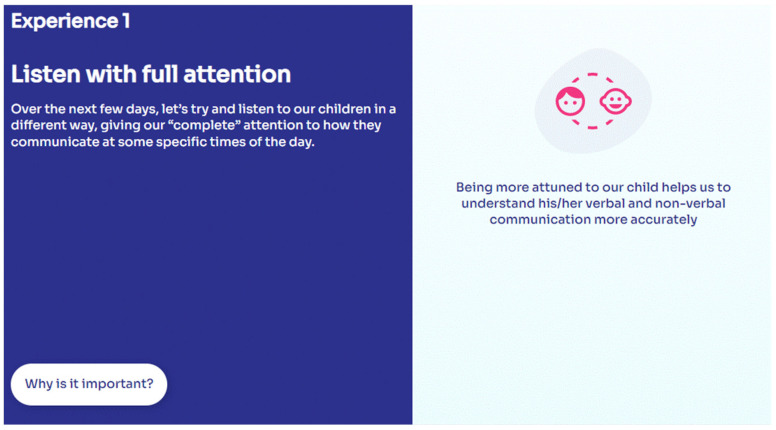
MinUTo app, an example of the section “Why is it important?” for the experience “Listening with full attention”.

**Figure 3 ijerph-19-07564-f003:**
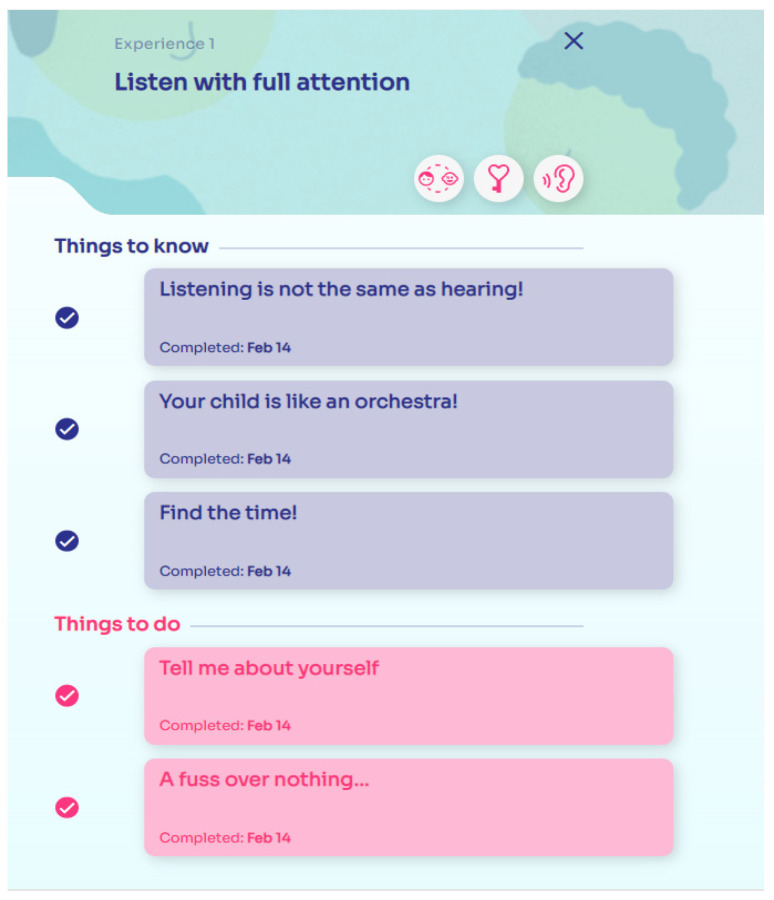
MinUTo app, examples of the sections “Things to know” and “Things to do” for the Experience “Listening with full attention”.

**Figure 4 ijerph-19-07564-f004:**
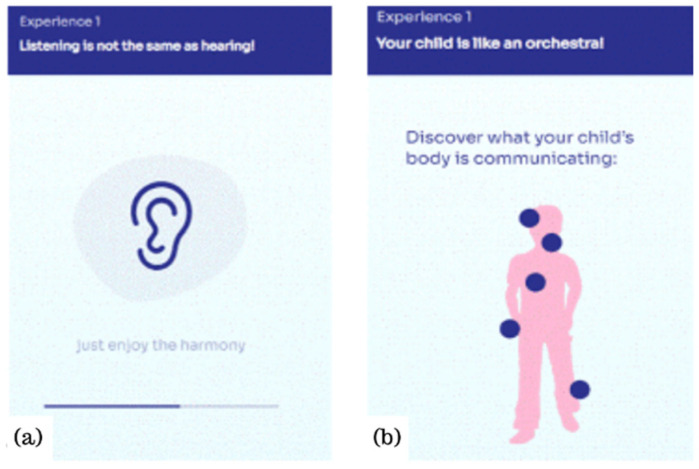
Two examples of interactive activities within the section “Things to know” for the Experience “Listening with full attention”. The first activity (**a**) is “Listening is not the same as hearing”. The second activity (**b**) is “Your Child is like an orchestra!”.

**Figure 5 ijerph-19-07564-f005:**
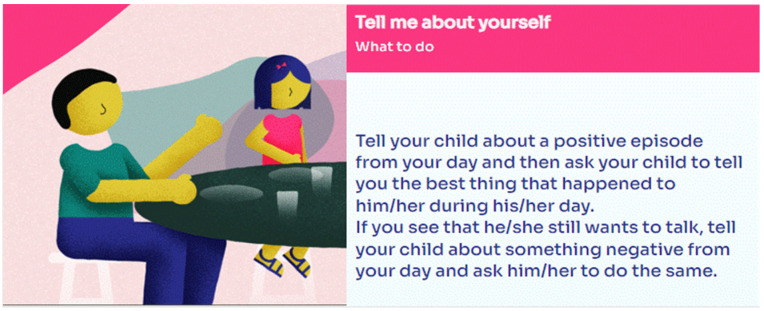
Example of activity within the section “Things to Do” for the experience “Listening with full attention”.

## Data Availability

Not applicable.

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
