# Peer review of "Mindful Parenting Intervention MinUTo App for Parents of Preschool Children: Study Protocol of a Randomised Controlled Trial"

_ijerph, 2022, doi:10.3390/ijerph19137564_

Round 1

Reviewer 1 Report

The article presents research with an interesting sample on a topic that is little addressed in the field of education with young children.

The method is coherent and rigorous, and provides information that situates the reader and supports replication in other contexts.

The results are explanatory and are developed coherently with the stated objectives (although they present more information than necessary).

The conclusions can be improved with respect to the hypotheses raised on pages 3 and 4. It would be good to take them up again in this section.

Author Response

  1. The article presents research with an interesting sample on a topic that is little addressed in the field of education with young children. The method is coherent and rigorous, and provides information that situates the reader and supports replication in other contexts. The results are explanatory and are developed coherently with the stated objectives (although they present more information than necessary). 

We thank the Reviewer for appreciating the topic of our manuscript as well as the methods and expected results. 

2. The conclusions can be improved with respect to the hypotheses raised on pages 3 and 4. It would be good to take them up again in this section. 

We agree with the Reviewer and insert a summary of the hypothesis in the conclusions (see p. 13), helping the reader to understand the Study Protocol's implications.     

Reviewer 2 Report

The article develops a subject with some interest and little studied. In general, it is written in a clear and coherent way.
The theme is based on current and credible bibliographical references, and is well presented. The authors try to justify the reason of the study and detail the design of the investigation. From the methodological point of view a doubt arises that should be better explained in the text:
1- The authors refer the use of two research instruments translated from Dutch, but their translation is not enough...it will be necessary to proceed to their cultural adaptation and make their validation for a new population. However, there is no reference to this process.
2- the randomisation process also deserves better explanation.

Finally, it is true that it is an interesting research protocol, but of course, being only a project, the absence of results reduces the interest of the article.

Author Response

  1. The article develops a subject with some interest and little studied. In general, it is written in a clear and coherent way. The theme is based on current and credible bibliographical references, and is well presented. The authors try to justify the reason of the study and detail the design of the investigation. 

We thank the Reviewer for appreciating the manuscript concerning the topic, references, and design. 

2. From the methodological point of view a doubt arises that should be better explained in the text: 1- The authors refer the use of two research instruments translated from Dutch, but their translation is not enough...it will be necessary to proceed to their cultural adaptation and make their validation for a new population. However, there is no reference to this process. 

We agree with the Reviewer that the back-translation is not enough for the process of validation in another language and culture. For this reason, we have included more information in the paragraph on Planned Statistical Analysis, suggesting the importance of carrying out Confirmatory Factor Analyses (see p. 12). We have also added a brief comment in the limitations paragraph (see p.13).     

3. The randomisation process also deserves better explanation. 

We have added more information on the randomization process (see p. 4).  

 4. Finally, it is true that it is an interesting research protocol, but of course, being only a project, the absence of results reduces the interest of the article. 

We agree with the Reviewer and hope we can publish interesting results on MinUTo App intervention in the following years. However, study protocols are necessary to specify the intervention studies' research plan.  

Round 2

Reviewer 2 Report

The authors made the suggested changes.